# An Entailment Tree Generation Approach for Multimodal Multi-Hop Question Answering with Mixture-of-Experts and Iterative Feedback Mechanism

## ABSTRACT

With the rise of large-scale language models(LLMs), it is currently popular and effective to convert multimodal information into text descriptions for multimodal multi-hop question answering. However, we argue that the current methods of multi-modal multi-hop question answering still mainly face two challenges: 1) The retrieved evidence containing a large amount of redundant information, inevitably leads to a significant drop in performance due to irrelevant information misleading the prediction. 2) The reasoning process without interpretable reasoning steps makes the model difficult to discover the logical errors for handling complex questions. To solve these problems, we propose a unified LLMs-based approach but wihout heavily relying on them due to the LLM's potential errors, and innovatively treat multimodal multi-hop question answering as a joint entailment tree generation and question answering problem. Specifically, we design a multi-task learning framework with a focus on facilitating common knowledge sharing across interpretability and prediction tasks while preventing task-specific errors from interfering with each other via mixture of experts. Afterward, we design an iterative feedback mechanism to further enhance both tasks by feeding back the results of the joint training to the LLM for regenerating entailment trees, aiming to iteratively refine the potential answer. Notably, our method has **won the first place** in the official leaderboards of WebQA (since April 10, 2024), and achieving competitive results on MultimodalQA.

## CCS CONCEPTS

• **Computing methodologies** → *Reasoning about belief and knowledge*.

## KEYWORDS

Multimodal Multi-Hop Question Answering, Entailment Tree, knowledge reasoning

## 1 INTRODUCTION

Multimodal multi-hop question answering (MMQA)[3] is a complex task that involves multiple input sources such as text, tables, and images. It requires reasoning through different modalities to

*ACM MM, 2024, Melbourne, Australia*

© 2024 Copyright held by the owner/author(s). Publication rights licensed to ACM.
ACM ISBN 978-x-xxxx-xxxx-x/YY/MM
https://doi.org/10.1145/nnnnnnn.nnnnnnn

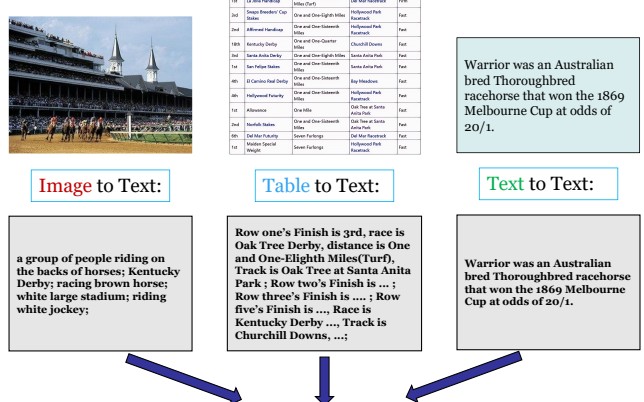

**Figure 1: Examples of current methods that converting multimodal information to text on the MultiModalQA dataset. In the figure, much redundant multimodal information has been converted into text(black font). Meanwhile, the key information has a strong logical relationship with each other, and current methods also have not utilized this relationship.**

generate accurate and complete answers. Currently, most multimodal multi-hop question answering methods adopt the approach of converting multimodal information into textual descriptions (by transforming images through image caption models, and tables through natural language descriptions), and then using large-scale language models (LLMs) to generate answers [8, 11, 26, 27]. The salient advantage of this method is that it can leverage the powerful language understanding and generation capabilities of LLMs, as well as the interpretability of textual descriptions. However, this method indiscriminately converts all multimodal information into textual descriptions, inevitably producing a large amount of redundant information. As shown in Figure 1, the current general method in the field of multimodal question answering only requires a small amount of key information to answer the multi-hop question. For instance, to find the "churchill down truck", only the fifth row of the table is needed. The other information in the table is not helpful to answer the question. Similarly, in the picture, only the "racing brown horse" needs to be paid attention to. The other entities are of little help to answer the question.

However, the previous method generates too much redundant textual information, which may mislead the model into generating incorrect answers. More importantly, there are an inherent reasoning relationships between these key pieces of information, which

has been ignored by the current methods for interpreting and utilizing this logical relationship between them, avoiding the logical and factual errors when dealing with complex questions [17].

Different from previous methods with redundancy and having no interpretability, we innovatively treat multimodal multi-hop question answering as a joint entailment tree generation and question answering problem. To the best of our knowledge, this is the first work *introducing entailment tree generation from a mutual-beneficial perspective, bridging small model and large model generation* in multimodal multi-hop question answering.

In the field of Natural Language Processing(NLP), entailment tree generation is an important sub-task in the question-answering task [6, 10, 20] while it has not yet been applied in the multimodal domain. The reason is that it is nontrivial to directly apply the existing method in NLP to our cases for benefiting the answering beyond the interpretability. During our attempt to directly transfer entailment tree generation methods to the multimodal multi-hop question-answering domain, we discovered that the current entailment tree generation methods *have very low accuracy* in generating entailment trees. *Even predicting the structure of entailment trees yields similarly low accuracy.* Therefore, we propose a new LLMs-based method that introduces the task of entailment tree generation into the multimodal multi-hop question answering task, to address both prediction and interpretability problems, and allows the introduced entailment tree to be iteratively refined.

Although LLMs show promising performance in many tasks, we still have observed that *the entailment trees generated by LLM mainly contain two types of errors*: 1) incorrect selection of leaf nodes. 2) incorrect structure of the entailment tree. Therefore, we propose the LLM and smaller model interactively based framework to carry out the fact retrieval generation task and the question-answering task. We use multi-task learning with smaller models, and use an iterative feedback mechanism to re-predict the structure of the entailment tree based on the leaf nodes and the answers predicted by the small model. Inspired by the idea of [9, 22, 23] , to facilitate mutual enhancement among the small models' multi-task learning, we employ a shared multi-task mixture-of-experts model, allowing interactions between the fact selection and supervised QA tasks as guidance for LLM.

Specifically, in entailment tree initialization stage, we iteratively use large-scale language models[2] to decompose an existing multi-hop question into sub-questions that need to be solved, and completes them based on existing evidence (question, answer) for as facts to construct a fact base. For entailment tree generation [6], since even predicting the structure of the entailment tree can only achieve a very low accuracy, which is much harder for our case, we propose to first generate entailment tree structure without the details, using the existing method of entailment tree generation [6] in NLP, then use large models to continuously fill in the values of the missing intermediate nodes in the entailment tree.

*Different from those in previous entailment tree generation tasks, where the leaf nodes and answers were provided simultaneously*[6, 10, 20], for multi-modal multi-hop question answering tasks, the answers are not visible during testing, so our definition of entailment tree generation tasks is different from the past[6]. When we build the entailment tree, the input is a set of leaf nodes (facts from the fact base) and a question as hypothesis, while previous methods

require inputting a set of leaf nodes and answer. After initializing the entailment tree in our method, both the leaf nodes and intermediate nodes are filled (the intermediate nodes are generated by the LLM through the collection of their child nodes), and we do not predict the final answer during the initialization phase of the entailment tree, but *predict the answer in the second stage.* For a detailed definition of our entailment tree generation, please refer to section 3.1.2.

Our method introduces entailment tree generation into the field of multi-modal multi-hop question answering, filters facts by generating entailment tree, models logical relationships between different modalities, eliminates irrelevant information in multi-modal contexts, and maintains logical consistency.

We conduct experiments on two public MMQA datasets, namely WebQA [3] and MultiModalQA [21]. We use accuracy, F1 score and reasoning path quality as evaluation metrics. Our experimental results show that our method achieve sota result on WebQA dataset. We also show the entailment trees generated by our method, demonstrating the effectiveness and explainability of our method. To the best of our knowledge, this is the first attempt to improve Multimodal Multi-Hop QA that uses entailment trees to constrain the process of converting multimodal information into text. The main contributions of our paper are as follows:

- By constructing a fact base, we reduce information redundancy, and use the fact base to build an entailment tree to generate explicit reasoning steps, which assist the model in generating more accurate answers.
- We introduce entailment tree generation into multi-modal multi-hop question answering. In order to correct potential errors in the entailment tree generated by the LLM, we proposed a multi-task mixture-of-experts model and iterative feedback mechanism.
- We achieve state-of-the-art results on WebQA dataset, and achieve competitive results on MultimodalQA dataset, demonstrating the effectiveness of our method.

## 2 RELATED WORK

**Multimodal Multi-Hop Question Answering** Multimodal multi-hop question answering is a task based on multimodal question answering, but it requires multi-hop reasoning to generate the final answer. VQA [1] is first proposed to answer questions from visual-only inputs. Later, WebQA [3] and MultimodalQA [21] require integrating information across free text, images, or semi-structured tables, to answer multi-hop reasoning question. To address the challenge of finding answers from multiple sources of information, MuRAG [5] designs a multi-modal transformer architecture to accept both text and image feature inputs, and builds a million-scale dataset for pretraining the model. [17, 17, 25, 27] unified multimodal information into text using image caption model and table linearization method, they proposed a new multimodal question answering paradigm, but there is no restriction during transfer, resulting a lot of information redundancy and affecting the performance of the model. Also, there are many recent works on multimodal question answering using large models. [11] train an image caption model to generate image caption for gpt-3 to understand images then generate responses; [15] use multimodal large model LLaVA to generate

more accurate image caption, then construct different in-context learning templates according to each modalities, enabling GPT-3 to leverage its powerful performance in this task. Both approaches need to generate image caption for the large language model to understand question, but there are no conditional restrictions during the image caption generation stage; or when generating image captions directly based on multi-hop questions, the questions contain information cannot be asked by a single image, which causes errors during the image caption generation stage.

For the above problems we found, we proposed an approach to filtered redundant information through the logical structure of the entailment tree, ensure the simplicity of information and the rationality of reasoning.

**Entailment Tree Generation** The task of entailment tree generation currently serves NLP question answering systems primarily. [6] introduce EntailmentBank, a dataset specifically designed for the task of entailment tree generation. Each multi-step entailment tree in EntailmentBank serves as an explanation, clearly demonstrating the reasoning process behind a hypothesis based. Recent methods[10, 14, 20] have presented multi-step generation approaches, which iteratively select premise facts and generate intermediate conclusions.

At present, the all correct score(only if all of the leaves, steps, and intermediates are all correct) of entailment tree structure generation in the field of NLP is very low(2.9% in full corpus) and cannot be directly applied to other fields; however, in the current multimodal question answering datasets, the questions are relatively simple. According to the statistics of two datasets [3, 21], for the MultimodalQA dataset, the proportion of complex questions (with reasoning hops greater than or equal to 3) is only 11.3%. After removing the simple multiple comparison questions, the proportion of complex questions only accounts for 1% of the total dataset. For the WebQA dataset, the proportion of complex questions (with reasoning hops greater than or equal to 3) is only 1%. These two datasets are currently the most complex in multimodal multi-hop question answering, suitable for evaluating our methods.

**Multi-Task Mixture-of-Experts** Recent rumors suggest that GPT-4's internal structure employs a mixture-of-experts (MoE) approach, which has been influential in the development of large-scale models[12, 23, 28]. The use of MoE as an architectural foundation has become prevalent in recent large models, propelling the advancement of both the models themselves and the MoE concept.

Moreover, multi-task MoE models have seen significant development prior to their integration into large-scale models. [18] propose a multi-layer gated network based on different tasks, allowing each task to have its independent experts, thereby enabling the model to better capture the inter-task correlations. Additionally, based on the previous method, [22] propose a method which retains the shared experts, allowing for interaction between different experts. [9] devise a task-aware gating mechanism within sparse MoEs to route the input (tokens from different tasks) to specialized experts conditioned on the task.

We combine the methods of multi-task MoE from previous research with the current MoE training approaches based on large models, enabling the multi-task MoE model to be suitable for multi-task learning with data generated by large-scale models.

# 3 METHOD

As shown in Figure 2, our method is divided into two stages: (a) entailment tree initialization stage and (b) iterative mixture-of-experts optimization stage. The goal of the first stage is to initialize the entailment tree for the use of small model in assisting with answering and providing interpretability. We decompose the original question to build a fact base, and use LLM to initialize the structure of the entailment tree based on the fact base. The goal of the second stage is to correct the leaf node and structural errors in the initialized entailment tree through joint learning of fact retrieval generation task and question answering task, and to iteratively optimize the entailment tree through the feedback of the results of joint learning to the LLM.

## 3.1 Entailment Tree Initialization Stage

*3.1.1 Fact Base Construction.* In the fact base construction module, we need to decompose the multi-hop question into several sub-questions based on different evidence and process them differently according to the modality of the corresponding evidence.

**Decompose Multi-Hop Question** First, we need to retrieve the multimodal evidence required to answer the multi-hop question. However, since our method generates image captions by decomposing the question, we use the global image caption and image attribute features to retrieve evidences according to the method in [27], and finally retrieve the required multimodal evidence set E:

$$E = \text{BERT}_{retri}(Text_{evidence}) = [E_1, E_2, ..., E_n], \quad (1)$$

where "n" represents the number of evidence in the evidence set. '$Text_{evidence}$' refers to all the evidence obtained after converting 'images, text, tables' into text, which is referred to as '$Text_{evidence}$'. After obtaining all the retrieved multimodal evidence $E = [E_1, E_2, ..., E_n]$, we prompt GPT-3.5 to decompose the original question based on all the evidence $E$ and generate n sub question $q^s$ with their corresponding evidence. Suppose that the k-th question of $q^s$ has L tokens, denoted as $q_k^s = (y_k^1, y_k^2, ..., y_k^L)$, the decoding process can be formulated as:

$$y_k^l = \text{argmax}_{LLM}(E_k, y_k^l | y_k^{<l}; p_q, q, E) \quad (2)$$

where $p_q$ is the instruction prompt. The outline of the prompt $p_q$ for LLM is as shown in Figure 3:

After obtaining all the decomposed sub-questions based on each evidence, we process them differently according to the modality of the corresponding evidence and convert them into facts stored in the fact base.

**Image Fact** For the image modality, we further decompose each sub-question into atomic questions $q^i$ using GPT-3.5. Suppose that the r-th atomic question of $q^i$ decomposed from $q_k^s$ has $L^i$ tokens, denoted as $q_k^i = (y_k^1, y_k^2, ..., y_k^{L^i})$, the decoding process can be formulated as:

$$y_r^{l^i} = \text{argmax}_{LLM}(y_r^{l^i} | y_r^{<l^i}; p_q, q_k^s, E_k) \quad (3)$$

Then, we input the decomposed atomic question $q^i$ and corresponding image $E_i m$ into the VQA model to obtain answers. We use LLaVA-1.5[13] as our VQA model. it uses Clip[19] as the visual feature extractor:

$$Feature_{img} = \text{CLIP}(E_k) \quad (4)$$

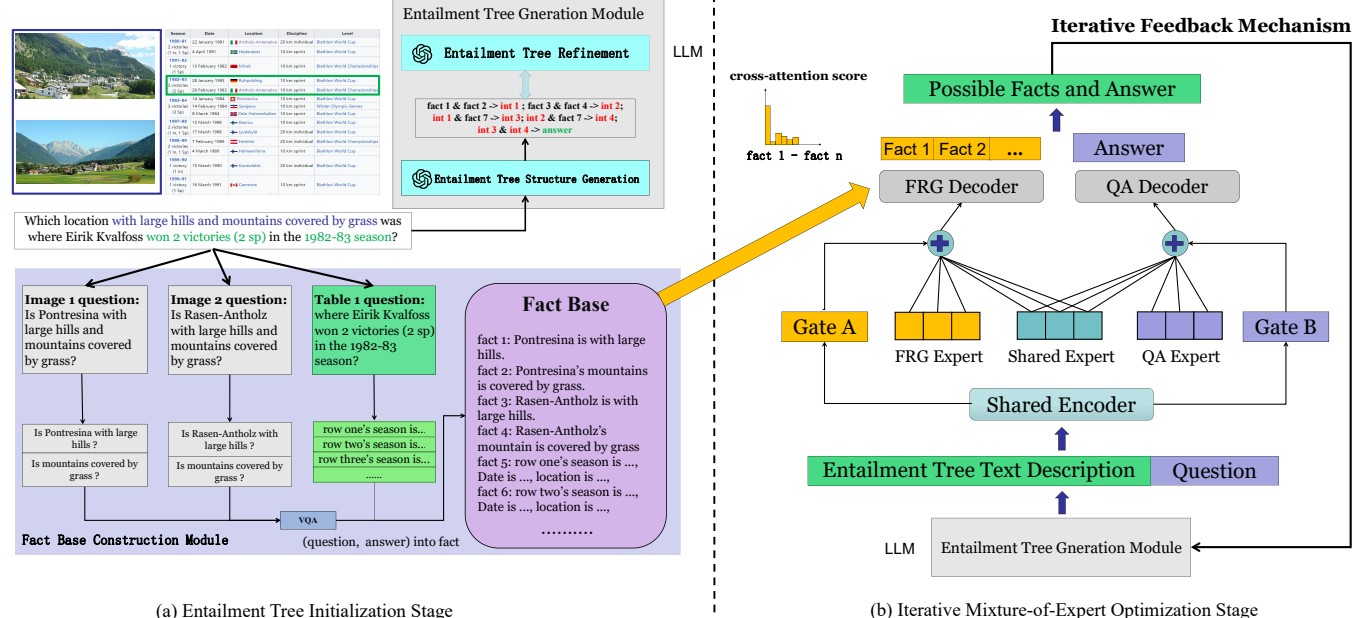

(a) Entailment Tree Initialization Stage

(b) Iterative Mixture-of-Expert Optimization Stage

**Figure 2: In our proposed (a) Entailment Tree Initialization Stage, we build a fact base by decomposing multi-hop questions and initialize an entailment tree using GPT-3.5. In our proposed (b) Iterative Mixture-of-Experts Optimization Stage, we convert the initialized tree into text, concatenate it with the origin question, and then input into shared encoder, and using two separate gates to select experts, then use two decoders for fact retrieval generation and question answering, where the FRG Decoder denoted as the Fact Retrieval Generation Decoder, and the QA Decoder denoted as Question Answering Decoder. After these two tasks, we convert the retrieved facts and answer back into text for reference in regenerating the tree structure using Iterative Feedback Mechanism.**

```
/* Instruction for the decompose question task */
Please decompose the TARGET-QUESTION into K sub questions:
sub questions:
/* n in-context examples */
TARGET-QUESTION: q1 \n
EVIDENCES:
<Image Evidence 1> Evidence image \n
<Table Evidence 1> Evidence table \n
<Text Evidence 1> Evidence text \n
Sub-Questions:
<Image Evidence 1>: sub-question for image evidence 1 \n
```

**Figure 3: The prompt template of decompose multi-hop question**

Suppose that the r-th atomic question's answer $a^r$ has $L^a$ tokens, denoted as $a_k^r = (y_k^1, y_k^2, ..., y_k^L)$, the decoding process can be formulated as:

$$y_r^{l^a} = \text{argmax}_{VQA}(y_r^{l^a}|y_r^{<l^a}; q_k^i, Feature_{img}) \qquad (5)$$

Then we use GPT-3.5 to refine the obtained (atomic question, answer) pairs into facts stored in the fact base.

**Table Fact** After we decompose the origin question, and obtained sub-question $q^t b$ for table evidences, we use [27] which use simple natural language templates to transform tables into sentences that

sound natural to humans. As an example, we can turn the table into a sentence by arranging the cells in a linear fashion, like this: "row one's seaon is ..., Date is ..., location is ...", and then feed into GPT-3.5 to obtain (sub-question,answer) pairs into facts stored in the fact base. Suppose that the t-th table's sub-question $q^t b_t$ has $L^{tb}$ tokens, denoted as $q^t b_t = (y_t^1, y_t^2, ..., y_t^L)$, the decoding process can be formulated as:

$$y_t^{l^{tb}} = \text{argmax}_{LLM}(y_t^{l^{tb}}|y_r^{<l^{tb}}; p_{tb}, q_t^{tb}, E_k) \qquad (6)$$

### 3.1.2 Entailment Tree Generation.

*Definition 3.1.* The entailment tree generation task input consists of a corpus of premises C (facts from fact base) and a hypothesis h (original question). The objective is to generate an entailment tree T that explains the hypothesis h by using a subset of the premises in C as building blocks. Entailment trees are represented as a tuple $T = (h, L, E, S)$, where leaf nodes $l_i \in L$ are retrieved from the corpus (i.e. $L \subseteq C$), internal tree nodes $e_i \in E$ are intermediate conclusions (new sentences not present in corpus C, note that intermediate conclusions are generated by LLM), and $s_i \in S$ is a list of entailment steps that can explain the hypothesis h, which is always the tree root and the final conclusion.

**Entailment Tree Structure Generation** The main function of the entailment tree structure generation module is to select leaf nodes and generate the entailment tree structure. At this stage, we

use predefined symbols for the entailment tree structure generation task which introduced by [6], where the facts in the fact base serve as a set of possible leaf nodes, and the original question serves as the conclusion. The entailment tree structure is generated using GPT-3.5, where the facts are the leaf nodes, and several leaf nodes are combined to form intermediate nodes. In the entailment tree structure generation module, only facts and the original question are given, and the intermediate nodes generated do not contain any specific information. In the initialization phase of the entailment tree, we do not directly predict the answer. Instead, we continuously fill the entailment tree with leaf nodes by predicting the structure of the entailment tree, and constantly improve the corresponding intermediate nodes through the combination of leaf nodes.

After predicting the structure of the entailment tree based on the question and perfecting the entailment tree, since the root node (question) does not contain the answer, we use the "answer" placeholder to replace the root node and input it into the second phase to perfect the entailment tree.

We use the symbol "&" to denote "and", and "->" to denote "entails", Suppose that the j-th origin question $s_j$ and fact base $FB = [fact_1, fact_2, ..., fact_m]$ are input into GPT-3.5 and generate entailment tree structure $t^s$ which has $L^t$ tokens, denoted as $t_j^s = (y_j^1, y_j^2, ..., y_j^L)$, the decoding process can be formulated as:

$$y_j^{l^t} = \text{argmax}_{LLM}(y_j^{l^t} | y_j^{<l^t}; p_t, q_j, FB) \qquad (7)$$

where $p_t$ is the instruction prompt. The template of the prompt $p_t$ for LLM is as follows:

```
/* Instruction for the decompose question task */
Please generate entailment tree according to the given facts and target-question.\n
/* n in-context examples */
TARGET-QUESTION: origin question q \n
Facts: fact 1: ... , fact 2: ... , ...
Entailment Tree: fact & fact2 -> int1; ...
```

**Figure 4: The prompt template of entailment tree structure generation**

**Entailment Tree Refinement** The entailment tree refinement module mainly completes the intermediate nodes span in the already generated entailment tree structure.

The algorithm we use is shown below: The meaning of "Split the entailment tree structure into multiple sub-trees set T" is that we start from the root node of the constructed entailment tree structure, continuously obtain subtrees composed of non-leaf nodes $N_r$ and their child nodes $N_c$ ($Depth(N_c) = Depth(N_r) + 1$), and continuously add them to the set to form a set of subtrees T. we split the set of all subtrees in the entailment tree structure. If the number of subtrees in the subtree set is greater than 1, we input all the subtrees one by one into the large model in order and deduce the root (intermediate node) of each subtree based on the leaf nodes (facts). Finally, we merge them into a complete entailment tree $Tree_{initial}$.

## 3.2 Iterative Mixture-of-Experts Optimization Stage

Due to the possible leaf node selection errors and entailment tree structural errors in the previously mentioned initialized entailment

**Algorithm 1** Algorithm of Entailment Tree Refinement.

---
**Require:** Fact Base, Entailment Tree Structure.
 1: Split the entailment tree structure into multiple sub-trees set $T$.

 2: **if** $length(T) > 1$ **then**
 3:     **for** $i = 0$ to $length(T)$ **do**
 4:         Split sub-tree $T_i$ into node set $N$
 5:         **for** $j = 0$ to $length(N)$ **do**
 6:             **if** $N_j$ in Fact Base **then**
 7:                 Replace $N_j$ with $N_j$ in Fact Base
 8:             **else**
 9:                 Inference intermediate node with other node using GPT-3.5
10:             **end if**
11:         **end for**
12:     **end for**
13:     Complete Entailment Tree = Concat([k for k in T])
14: **else**
15:     Complete Entailment Tree = T
16: **end if**

---

tree, we use a hybrid expert model to jointly learn the fact retrieval generation task (retrieving leaf nodes) and question answering tasks, and correct the entailment tree structure through an iterative feedback mechanism.

### 3.2.1 Jointly Learning of Fact Retrieval Generation And Question Answering.
First, we use T5 encoder to extract the features of each fact in the fact base as $F_{fact} = [f_{[mean]}^1, f_{[mean]}^2, ..., f_{[mean]}^n]$, Afterwards, we convert the entailment tree into a natural language description denote as $Tree_{initial}^{text}$ and input $Tree_{initial}^{text}$ into the shared encoder of T5, and then concat it with origin question $q$ to obtain the final feature $F_{et}$:

$$F_{et} = SharedEncoder(Concat([Tree_{initial}^{text}, q])) \qquad (8)$$

Once we have obtained the features, we input $F_{et}$ into the Mixture-of-Experts model.

**Multi-Task Mixture-of-Experts** our mixture-of-experts model primarily consists of three parts: two gating networks, two task-specific expert networks, and one shared expert network. The task-specific expert networks are dedicated to the tasks of fact retrieval generation and question answering, respectively, while the shared expert network can be utilized for both tasks. The gating network is responsible for selecting the appropriate experts. Specifically, gating network A selects from the fact retrieval generation task and shared experts, while gating network B selects from the question answering task and shared experts.

**Top-2 Selection**. According to the formulation above, when g(·) is a sparse vector, only part of the experts would be activated and updated by back-propagation during training. We set the gating layer as a top-K selection as:

$$g(F_{et}) = TopK(softmax(f(F_{et}))) \qquad (9)$$

where $f(F_{et})$ is routing linear transformation $R^D \rightarrow R^E$.

**Token-choice Routing** We generally follow [23] for our routing design to ensure training stability. Given E trainable experts and

input representation $F_{et} \in R^{len^{et}*D}$, the output of MoE model can be formulated as:

$$MoE(F_{et}) = \sum_{i=1}^{n} g(F_{et})_i e^i(F_{et}) \tag{10}$$

where $e^i(F_{et})$ is a non-linear transformation $R^D \rightarrow R^D$ of the ith expert, and $g(F_{et})_i$ is the i th element of the output of the trainable router $g(F_{et})$, a non-linear mapping $R^D \rightarrow R^E$. Usually, both $e(F_{et})$ and $g(F_{et})$ are parameterized by neural networks. Please note each expert is an FFN layer instead of a complete Transformer model in most MoE-based Transformer models, including ours.

After we have selected the experts for the fact retrieval generation task and the question answering task through the gated network, we add $MoE(F_{et})$ with $F_{et}$ to obtain the final MoE output:

$$MoE_f = add(MoE(F_{et}), F_{et}) \tag{11}$$

then we input $MoE_f$ into the decoder of their respective tasks.

**Decoders for Multi-Tasks** We handle the two decoders differently. The FRG Decoder performs cross-attention with all fact features and the entailment tree description passed through the Shared Encoder, while the QA Decoder only performs cross-attention with the entailment tree description passed through the Shared Encoder. The reason for doing this is that we hope the QA model can get the answer based solely on the entailment tree. If it cannot, then optimize the entailment tree through the FRG model and the Iterative Feedback Mechanism until the QA model can answer the question based solely on the entailment tree.

The decoder for fact retrieval generation $Decoder_{frg}$ performs cross attention with the facts feature $F_f act$ and shared encoder output $F_{et}$:

$$w_t, g_t = CrossAttention(CrossAttention(Decoder(q), MoE_f), F_{fact}) \tag{12}$$

where $w_t$ denotes the cross-attention weights at time step t. then the decoder $Decoder_{frg}$ begins to retrieval at time step $|Q|$, i.e., the length of the question $q$, and then we utilize cross-entropy loss for it:

$$L_{frg} = -\frac{1}{M} \sum_{t=0}^{M} \log \frac{\exp(\alpha_{t,t+})}{\sum_{i=1}^{n} \exp(\alpha_{t,i})} \tag{13}$$

where $\alpha_{t,i}$ denotes the cross-attention scores of the i-th fact at time step t, $\alpha_{t,t+}$ denotes the score of the target source(The fact index sequence extracted from the entailment tree generated by LLM) at time step t, M is the number of retrieval steps.

The decoder for the question answering task $Decoder_{qa}$ performs cross attention with the overall features $MoE_f$:

$$g_q = softmax(crossattention(Decoder(q), MoE_f)) \tag{14}$$

the decoder $Decoder_{qa}$ begins to retrieval at time step $|Q|$, i.e., the length of the question $q$, and then we utilize cross-entropy loss for it:

$$L_{qa} = \sum_{t=0}^{|A|} -\log P_t(a_i | MoE_f, a_{<t}) \tag{15}$$

The overall loss function is as follows:

$$L = L_{frg} + L_{qa} \tag{16}$$

**3.2.2 Iterative Feedback Mechanism.** After we complete the tasks of fact retrieval generation and question answering, we first replace the fact index obtained from the fact retrieval generation task in the fact base with the corresponding facts. Then, we concatenate it with the final answer and input it into GPT-3.5 as additional information to correct the entailment tree and conduct a second round of training. We add a prompt $p_i fm$ to the original prompt template which we use to generate the entailment tree structure: "Given the following potentially relevant facts and the potentially correct answer, please generate entailment tree in n words. Facts:f Answer:a Question:q"

---

**Algorithm 2** Algorithm of Iterative Feedback Mechanism.

---

**Require:** facts set $f$ from fact retrieval generation task, answer $a$ from question answering task.
1: iterative number $k = n$; $f^i = f$; $a^i = a$
2: **for** $i = 0$ to $k$ **do**
3:     Fill the prompt with $f^i$ and $a^i$ to get a complete prompt $p_c^i$.
4:     Concat $p_c^i$ with $p_t$(shown in Figure 4)
5:     Continue the processing of stage 1 to get a new entailment tree $T^i$.
6:     Continue with the steps of stage 2 through $T^i$, and get $f_{new}^i$ and $a_{new}^i$ respectively from the fact retrieval generation task and the question answering task.
7:     $f^i = f_{new}^i$; $a^i = a_{new}^i$
8: **end for**

---

## 4 EXPERIMENTS

### 4.1 Datasets

We conducted experiments on two of the most representative MMQA datasets: WebQA and MultimodalQA.

**WebQA** [3] is a multimodal and multi-hop question answering dataset that contains QA pairs that require one or two images and text snippets to answer. Each question has a set of distractors that the model must consider along with the correct clues to provide an answer. WebQA uses BARTScore to measure both the fluency and keyword accuracy of the answer denote as **QA-FL** and **QA-Acc** in Table 2. These two scores are multiplied together to obtain the **QA** score. The clue retrieval can be easily evaluated using **F1** score.

**MultimodalQA** [21] involves answering multi-hop complex questions by combining information from text, tables, and images. Each question also includes visual and text distractors. The performance is measured by F1 score at the word level and the Exact Match (**EM**) of the predicted answer.

### 4.2 Implementation Details

We conduct experiments on two datasets: WebQA, and MultimodalQA. The information source for WebQA includes both text and image modalities, while MultimodalQA focuses on text, images, and tables. For WebQA and MultimodalQA, a candidate clue list is given, and the model needs to find the most relevant clue to evaluate the accuracy of the clue retrieval. The backbone for retrieval is BERT[7]. We use LLaVA-1.5 as our VQA model, and use T5 as our FRG and QA model. We utilize the Transformers library and pretrained parameters from HuggingFace 4 and conduct experiments using 24G GPU

cards. Further, AdamW [16] is used as the optimization algorithm with a learning rate of 1e-4. The batch sizes for retrieval and qa are 32, 12.

## 4.3 Results

| Model | QA-FL | QA-Acc | QA |
|---|---|---|---|
| OFA-Cap + GPT-3 | 52.8 | 55.4 | 33.5 |
| PROMPTCAP + GPT-3 | 53.0 | 57.2 | 34.5 |
| **Our Method** | **60.1** | **77.2** | **47.1** |

**Table 1: Large language model Results on the WebQA validation set with oracle sources on image queries.**

We show our results of WebQA in Table 1-3. Table 1 shows all the methods that use large models on the webqa dataset. promptcap[11] trained an image caption model to generate image caption to let GPT-3 have more information about the images, however, the image captions generated by promptcap are too coarse-grained. Our method decomposes question by GPT-3.5 and filters the question to generate image caption, which let image caption model more focused on specific areas.

| Model | Retr | QA-FL | QA-Acc | QA |
|---|---|---|---|---|
| VLP [2022] | 0.69 | 0.43 | 0.37 | 0.23 |
| VLP + VinVL [2022] | 0.71 | 0.44 | 0.39 | 0.24 |
| MuRAG [2022] | 0.75 | 0.56 | 0.55 | 0.36 |
| SKURG [2023] | 0.88 | 0.56 | 0.57 | 0.38 |
| Solar [2023] | 0.89 | 0.61 | 0.59 | 0.41 |
| PERQA [2023] | **0.90** | 0.62 | 0.64 | 0.44 |
| **Our Method** | 0.89 | **0.68** | **0.73** | **0.54** |

**Table 2: WebQA official test-set[1] results indicated on leaderboard. we achieve the highest result on QA-FL, QA-ACC, QA score.**

| Model | QA-Acc | Retr |
|---|---|---|
| VitaminC | 57 | 84 |
| CMU ITL | 58 | 81 |
| HIT TMG | 58 | 89 |
| SDU | 69 | 86 |
| **Our Method** | **73** | **89** |

**Table 3: WebQA official test-set results on QA-Accuracy and Retrieve F1. Our method significantly exceeds other current methods in terms of QA-Accuracy.**

Table 2 shows all results on WebQA offcial test-set result, **MuRAG**[5] design a multimodal transformer, **SKURG**[24] design a entity fusion method, **solar**[27] unified multimodal into text. we also list the results of **VitaminC**, **CMU ITL**, **HIT TMG**, **SDU** on the EvalAI WebQA open leaderboard in Table 3.

[0] https://eval.ai/web/challenges/challenge-page/1255/leaderboard/3168

| Model | Single-Modal | | Mutli-Modal | | All | |
|---|---|---|---|---|---|---|
| | **EM** | **F1** | **EM** | **F1** | **EM** | **F1** |
| AR | 51.7 | 58.5 | 34.2 | 40.2 | 44.7 | 51.1 |
| ID | 51.6 | 58.4 | 44.6 | 51.2 | 48.8 | 55.5 |
| SKURG | 66.1 | 69.7 | 52.5 | 57.2 | 59.8 | 64.0 |
| PERQA | 69.7 | 74.1 | 54.7 | 60.3 | 62.8 | **67.8** |
| Solar | 69.7 | 74.8 | 55.5 | 65.4 | 59.8 | 66.1 |
| **Our Method** | **69.8** | **74.9** | **64.7** | **65.7** | **68.2** | 66.5 |

**Table 4: MultimodalQA dataset results.**

Also, as shown in Table 4, we surpass sota result on MultimodalQA in Single-Modal and Multi-modal set. This demonstrates that our method possesses superior reasoning abilities.

The WebQA and Multimodalqa dataset both requires measuring the accuracy of the final answer. Our method achieves much higher accuracy than other methods on the final answer, because in the first step of decomposing the question, GPT-3.5 can comprehensively decompose the question and match the final multimodal clues. This leads to a more focused generation of image descriptions based on the decomposed questions, providing reasoning steps for the final generation. Therefore, our method achieves a good result in terms of accuracy.

## 4.4 Ablation Study

| Model | Single-Modal | Mutli-Modal | All |
|---|---|---|---|
| | **EM** | **EM** | **EM** |
| **Our Method** | **69.8** | **64.7** | **68.2** |
| w/o decompose question | 68.6 | 55.2 | 64.8 |
| w/o LLaVA caption | 68.3 | 56.2 | 65.2 |
| w/o FRG | 67.2 | 58.4 | 65.2 |
| w/o MMOE | 69.8 | 63.4 | 67.3 |
| w/o IFM | 67.8 | 59.2 | 66.1 |

**Table 5: Ablation study on MultimodalQA dataset results. we denote Fact Retrieval Generation Module as FRG, Iterative Feedback Mechanism as IFM.**

In this experiment, we ablate the question decomposition, sub-question image caption modules, fact retrieval generation module, multi-task mixture-of-experts module, Iterative feedback module. When ablating the question decomposition module, we directly use the original question as input, and directly use the original question as the prompt for LLaVA to generate image captions; when ablating the sub-question image caption module and entailment tree generation modules, we directly concatenate the retrieved evidence and sub-questions and input them to GPT-3.5. The results of the ablation experiments are shown in Table 5. It can be seen that both question decomposition and the final sub-question image caption have a positive impact on the results, with the sub-question image description being particularly significant.

## 4.5 Case Study

As shown in Figure 5 and Figure 6, we present two case studies in the Multimodal datasets. Figure 5 shows the entailment tree generated

in the data sample of complex reasoning. Our method can generate effective entailment trees for a small amount of complex reasoning to guide the reasoning process. Figure 6 shows that when the first stage LLM generates an incorrect entailment tree (with errors in leaf node selection and structure), our proposed second stage can correct these errors through the joint learning of fact retrieval generation task and question answering task.

**Figure 5: Our Method can generate correct entailment tree according to the image, table and text.**

**Figure 6: When the entailment tree structure generated by the LLM is incorrect, or the selection of leaf nodes is wrong, our proposed Iterative Mixture-of-Expert Optimization Stage can correct the errors.**

| | p1 | p2 | p3 | p4 | p5 | p6 | p7 | p8 | p9 | p10 | Avg |
|---|---|---|---|---|---|---|---|---|---|---|---|
| q-True | 86 | 88 | 86 | 96 | 94 | 88 | 86 | 92 | 82 | 88 | 88.6 |
| r-True | 84 | 84 | 86 | 92 | 88 | 86 | 84 | 90 | 78 | 86 | 85.8 |

**Table 6: Quality Analysis and Explainability on WebQA dev set. p1-p10 is different evaluator, q-True is an indicator counting whether the question decomposition is correct. r-True is an indicator that counting whether the entailment tree's reasoning path is correct.**

## 4.6 Quality Analysis and Explainability

In this section, we show quality analysis of decomposed questions and explainability of final reasoning path, including the refined GPT-3.5 sub-questions and their specific image captions generated by LLaVA in our method. To evaluate the quality of question decomposition, we recruited 10 volunteers to conduct human evaluation. Each evaluator was randomly provided with 50 original questions, each origin question has corresponding sub-questions and reasoning path. We evaluate two indicators which are the accuracy of the decomposed question, the accuracy of the reasoning path. The accuracy of the decomposed question and the reasoning path is 89% and 86.2% respectively. We stipulate that when counting whether the question decomposition is correct, only when all the decomposed questions are correct can it be considered correct. When counting the explainable reasoning path, we require that the reasoning path be considered correct only when the evaluator thinks that the reasoning path can reason out the final answer.

## 5 CONCLUSION

We follow the current popular method [4, 8, 27], of unifying multimodal information into text information. In this text-driven multimodal paradigm, to the best of our knowledge, this is the first time that an explanatory improvement has been made from the perspective of entailment tree generation. In this paper, we construct an entailment tree through LLMs, and iteratively correct the entailment tree by proposing a multi-task MoE and iterative feedback mechanism. During the process, we can generate entailment trees based on the set of facts to assist the model in complex problem reasoning. Different from previous methods of entailment tree generation, our method not only has interpretability, but also helps to improve the accuracy of question answering. Our experiments demonstrate the potential of this approach.

## 6 LIMITATIONS

While our method has demonstrated its superior performance on two benchmarks, it still has several limitations. First, this method might not fully use the sub-question answers, for successively prompting next sub-question, although it may reduce error propagation. Second, the majority of data in current multimodal multi-hop question answering datasets does not entail complex reasoning. While the question decomposition module primarily contributes to performance enhancement when employing an entailment tree, we excel in handling a small fraction of complex problems by generating entailment trees, which can direct the model's reasoning path.

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
