# OpenReview forum: "An Entailment Tree Generation Approach for Multimodal Multi-Hop Question Answering with Mixture-of-Experts and Iterative Feedback Mechanism"
_acmmm.org/ACMMM/2024/Conference — MM2024 Poster_

### Official Review · Reviewer_N7Zp · 2024-05-15

**Rating:** 5
**Confidence:** 4

**Summary:**

This paper constructs a fact base to reduce information redundancy. It uses this fact base to build an entailment tree, which helps generate explicit reasoning steps, improving the model's accuracy in answering questions. The study introduces entailment tree generation into multi-modal, multi-hop question answering. To correct potential errors in the entailment tree generated by the LLM, the authors propose a multi-task mixture-of-experts model and an iterative feedback mechanism. This approach achievesSOTA performance on twoMMQA datasets.

**Strengths:**

Novelty: This paper is the first to introduce entailment tree generation from a mutual-beneficial perspective, effectively bridging small model and large model generation in multi-modal, multi-hop question answering. This novel idea adds significant value to the field.

Technical Approach: The use of a MoE architecture combined with an iterative feedback mechanism is innovative. This approach addresses potential errors in entailment tree generation by leveraging the strengths of multiple models, thereby enhancing overall accuracy.

Clarity: The paper is well-written and easy to follow. The authors provide clear explanations and logical reasoning, making the complex concepts accessible to the reader.

Adequate Evaluation: The authors demonstrate the effectiveness of their approach by achieving state-of-the-art performance on two MMQA datasets. This strong empirical evaluation justifies the practicality and robustness of their proposed method.

**Limitations:**

Parameter Details: The use of the MoE architecture likely increases the number of parameters significantly. It would be beneficial to include detailed information on the parameter count and compare it with baseline models to understand the trade-offs between performance gains and computational costs.

Table 5 Clarity: The paper lacks clarity in Table 5, as there is no corresponding line that specifically addresses the ablation of the entailment tree generation modules.

Iterative Feedback Mechanism: The paper does not specify how many iterations the iterative feedback mechanism performs. Including an ablation study on the number of iterations would provide valuable insights into the effectiveness and efficiency of this mechanism, helping to determine the optimal number of iterations needed for best performance.

**Suitability:**

3

---

### Official Review · Reviewer_mheX · 2024-05-20

**Rating:** 3
**Confidence:** 2

**Summary:**

This work focuses on multi-modal multi-hop question answering and treats it as a joint entailment tree generation and question answering problem. Firstly, the author design a multi-task learning framework to facilitate common knowledge sharing between interpretability and prediction tasks. Subsequently, an iterative feedback mechanism is designed to iteratively refine potential answers by feeding the joint training results back to the LLM to regenerate the potential tree.

**Strengths:**

- For the first time, the paper combines entailment tree generation and question answering tasks from the perspective of mutual benefit to bridge the generation of large language models and small language models.
- The authors also designed a multi-task mixture-of-expert model and an iterative feedback mechanism to iteratively refine the generated answers.
- Experimental results show that competitive results are achieved on WebQA and MultiModalQA datasets.

**Limitations:**

- This work provides a very detailed introduction to the method, but **lacks a comprehensive and in-depth analysis in the experimental results**. The analysis of this work in 4.3 Results and 4.4 Ablation Study is too general and lacks in-depth analysis of the proposed module. In addition, since the author only used two datasets for experiments, it is recommended that the author add ablation results on the WebQA dataset to better prove the role of the proposed module.
- The paper uses an iterative feedback mechanism, but **the paper does not report the iteration round k and the number of tokens consumed** (input tokens and output tokens) during the experiment, which are very important for evaluating the actual usability of the method.
- Is Table5 w/o MMOE _directly concatenate the retrieved evidence and sub-questions and input them to GPT-3.5_ method mentioned in th paper on line 802? If so, does this indicate that **the improvement of the model may come from the capabilities of GPT3.5 itself, rather than the modules proposed in the paper?** Because on the MultimodalQA dataset, the improvement of the method in the paper compared to w/oMMOE is not obvious compared to other modules.
- **Have the authors tried replacing GPT3.5 with an existing open source model?** For example, LLaMA series and mistral series. Can open source models also achieve better improvements?
- Some formula errors: line 404 $q_tb$, line 610 $F_fact$, line 645 $p_ifm$

**Suitability:**

3

---

### Official Review · Reviewer_v4G1 · 2024-06-01

**Rating:** 2
**Confidence:** 3

**Summary:**

This paper introduces an approach that combines multimodal multi-hop question answering with generative inference trees to improve the accuracy and interpretability of the reasoning process.
Through a hybrid expert model and iterative feedback mechanism, this method won the first place in the WebQA competition and achieved competitive results on MultimodalQA.

**Strengths:**

The advantages of this article include:

The joint method of inference tree generation and question answering is introduced to solve the problems of redundant information and unexplainable reasoning process in multimodal multi-hop question answering. By building an inference tree and filtering redundant information, the model can generate answers more accurately.

Combining the hybrid expert model and iterative feedback mechanism, it fully utilizes the advantages of large-scale language models and improves the performance and interpretability of the model.

**Limitations:**

The method of this paper is the same as other methods. In the end, it is also converted into text to infer the answer, but it is said that redundant information is filtered out. So the idea of ​​the article is confusing.

1. Is it necessary to use the tree generation method? The article does not compare some straight forward methods of filtering information.

2. Combining the tree generation and moe methods, is the performance improvement worth the extra computing cost and computing efficiency?

**Suitability:**

2

---

### Official Review · Reviewer_dbca · 2024-06-03

**Rating:** 3
**Confidence:** 3

**Summary:**

The paper addresses challenges in multimodal multi-hop question answering (MMQA) by proposing a novel approach that integrates entailment tree generation with question answering, enhancing both interpretability and accuracy. The paper proposes a unified approach that treats MMQA as a joint problem of entailment tree generation and question answering. This is achieved through a multi-task learning framework that shares common knowledge across tasks while preventing interference between them via a mixture of experts.

**Strengths:**

- The paper innovatively introduces entailment tree generation into MMQA, which is a novel and groundbreaking attempt in this field.
- The proposed framework is both effective and interpretable, achieving SOTA performance on the WebQA dataset.

**Limitations:**

A significant drawback of this paper is its disorganized presentation. Specifically:

1. Some figure captions lack punctuation, such as in Figure 4.
2. All equations are missing punctuation.
3. There are issues with citation formatting, failing to distinguish between cite and citet usage, as seen in line 229 and line 223 with double citations [17].
4. Inconsistent use of quotation marks, as observed in lines 318, 319, and 647.
5. Numerous capitalization errors, such as "it" instead of "It" in line 344, and "gpt-3" instead of "GPT-3" in line 230.
6. References to figures and tables do not enable proper hyperlink navigation.
7. The symbols used throughout the paper are inconsistent and confusing, as noted in lines 404 and 442. The authors should reorganize the use of symbols.
8. The example text in Figure 2(a) is very small and difficult to read.
9. Table 6 is not referenced in the text, potentially corresponding to Section 4.6. The percentages 89% and 86.2% in Section 4.6 are not clearly sourced.
10. The paper’s header has a title and conference information overlapping.
11. The footnote in the caption of Table 2 is incorrectly formatted as "0".
12. There should be a space before citations, such as in line 713.
13. Line 269 inappropriately uses the term "rumors" to support the author's point in an academic paper.

Although I find the paper's approach innovative, the poor presentation makes me inclined to reject it.

**Suitability:**

2

---

### Official Review · Reviewer_teL6 · 2024-06-10

**Rating:** 5
**Confidence:** 2

**Summary:**

The paper presents an innovative approach to multimodal multi-hop question answering (MMQA), a complex task that necessitates reasoning across different modalities such as text, images, and tables. The authors identify two key challenges in existing MMQA methods: the presence of redundant information and the lack of interpretable reasoning steps. To address these, the paper proposes a novel method that integrates entailment tree generation with a multi-task learning framework and an iterative feedback mechanism. This approach is particularly noteworthy for its application of a mixture-of-experts model to enhance both the interpretability and accuracy of the predictions.

**Strengths:**

1.The authors demonstrate the effectiveness of their approach through experiments on two public MMQA datasets, WebQA and MultiModalQA.
2. The introduction of entailment trees in the multimodal domain is novel. The use of a multi-task mixture-of-experts model to facilitate knowledge sharing and prevent task-specific errors is another innovative aspect of the proposed method.

**Limitations:**

1. Some typos in Line 21.

**Suitability:**

3

---

### Meta-Review · Area_Chair_R3Qs · 2024-07-07

**Recommendation:** Accept (Poster)
**Confidence:** 4

**Metareview:**

This paper innovatively addresses Multimodal Multi-Hop Question Answering with an entailment tree generation approach. The reviewers noted that the introduction of entailment tree generation to multimodal QA is both novel and interesting, potentially pioneering a new path in this research area. The proposed method has secured first place on the official leaderboard of WebQA (multimodal and multihop QA) and demonstrated strong performance on other multimodal QA datasets.

However, a significant drawback of this paper lies in its writing and presentation. As noted by Reviewer dbca and other reviewers, the paper contains some typos and formatting issues. Additionally, the AC observed that the language is somewhat verbose and clarity could be improved.

Despite these issues, the AC believes the novelty and effectiveness of this paper outweigh its drawbacks in presentation and recommends acceptance. The authors should carefully revise the paper according to Reviewer dbca's comments, proofread to identify and correct potential errors, and incorporate rebuttal discussions into the final version.